# Peptides from the Intestinal Tract of Breast Milk-Fed Infants Have Antimicrobial and Bifidogenic Activity

**DOI:** 10.3390/ijms22052377

**Published:** 2021-02-27

**Authors:** Robert L. Beverly, Prajna Woonnimani, Brian P. Scottoline, Jiraporn Lueangsakulthai, David C. Dallas

**Affiliations:** 1College of Public Health and Human Sciences, Oregon State University, Corvallis, OR 97331, USA; lueangsj@oregonstate.edu (J.L.); dave.dallas@oregonstate.edu (D.C.D.); 2College of Agricultural Sciences, Oregon State University, Corvallis, OR 97331, USA; woonnimp@oregonstate.edu; 3Division of Neonatology, School of Medicine, Oregon Health and Science University, Portland, OR 97239, USA; scottoli@ohsu.edu

**Keywords:** antimicrobial, bifidogenic, bioactive peptides, human milk, infants, intestine, peptidomics

## Abstract

For bioactive milk peptides to be relevant to infant health, they must be released by gastrointestinal proteolysis and resist further proteolysis until they reach their site of activity. The intestinal tract is the likeliest site for most bioactivities, but it is currently unknown whether bioactive milk peptides are present therein. The purpose of the present study was to identify antimicrobial and bifidogenic peptides in the infant intestinal tract. Milk peptides were extracted from infant intestinal samples, and the activities of the bulk peptide extracts were determined by measuring growth of *Escherichia coli*, *Staphylococcus aureus*, and *Bifidobacterium longum* spp. *infantis* after incubation with serial dilutions. The peptide profiles of active and inactive samples were determined by peptidomics analysis and compared to identify candidate peptides for bioactivity testing. We extracted peptides from 29 intestinal samples collected from 16 infants. Five samples had antimicrobial activity against *S. aureus* and six samples had bifidogenic activity for *B. infantis*. We narrowed down a list of 6645 milk peptides to 11 candidate peptides for synthesis, of which 6 fully inhibited *E. coli* and *S. aureus* growth at concentrations of 2500 and 3000 µg/mL. This study provides evidence for the potential bioactivity of milk peptides in the infant intestinal tract.

## 1. Introduction

Over 380,000 infants are born prematurely in the US each year [1]. Compared with infants born at full term, preterm infants are at heightened risk of developing infections such as sepsis [2] and necrotizing enterocolitis [3], with the infection risk increasing as gestational age at birth decreases [4,5,6]. Due to their reduced development time *in utero*, preterm infants are often born with an underdeveloped gastrointestinal (GI) tract and innate immunity (reduced gastric acidity, looser tight junctions, and dysbiotic microbiome) that leave them susceptible to pathogens [7,8,9]. The organisms most commonly responsible for systemic infections vary by hospital but typically include coagulase-negative *Staphylococcus*, group B *Streptococcus*, Gram-negative bacteria, and *Candida* [10,11,12,13,14]. The etiology of necrotizing enterocolitis is less well understood but is associated with aberrant colonization of the gut with a predominance of gammaproteobacteria and reduced commensal *Bifidobacteria* and *Bacteroidetes* species [15,16]. Though the causes and locations of infection are disparate among infants, the universal standard of care for risk reduction is early and dedicated enteral feeding with human milk, whether the mother’s or donor milk [17]. 

Human milk is the ideal source of nutrition for the preterm infant as it contains a variety of bioactive compounds that provide protection for the infant GI system. Immunoglobulins (Ig) and antimicrobial proteins such as lactoferrin and lysozyme inhibit bacterial growth [18,19,20,21], human milk oligosaccharides prevent pathogen adhesion [22,23] and act as a specific source of nutrition for commensal bacteria [24], and growth factors can facilitate the maturation of the intestinal epithelium [25,26]. Additional components of human milk potentially protective for the infant are milk peptides. Human milk proteins are exposed to a variety of proteases in the mammary gland and the infant GI system that initiate the degradation of the proteins into individual amino acids for infant nutrition. Protein digestion deactivates many of the functional proteins secreted in milk, but it also releases tens of thousands of peptides as an intermediary stage between the intact protein and its component amino acids [27]. 

Though the majority of these peptides are likely biologically inert, many have been identified with potential bioactive properties both similar to and distinct from their parent proteins through in vitro methods [28]. Of particular significance for infants are those with antimicrobial [29], immunomodulatory [30,31], and bifidogenic [32] properties that have the potential to provide additional immunological support as their GI system matures. However, the relevance of these peptides to infant health is dependent upon whether they are released during GI digestion and survive to their sites of activity. Several peptidomic studies have revealed that hundreds of bioactive peptides are released in human milk and the stomach of breast milk-fed infants [27,33]. Furthermore, bioactive milk peptides have been identified to survive to infant stool [34], but little is known about the presence or activity of bioactive peptides inside the infant intestinal tract. Bioactive peptides present in the intestinal tract have the highest potential to positively impact infant health, either by absorption into the infant’s circulation or by local activity on the intestinal cells and bacteria.

The aim of this cross-sectional study was to identify milk peptides in the intestinal fluids of breast milk-fed preterm and term infants and characterize them for antimicrobial and bifidogenic activity. Peptides were extracted from infant intestinal fluids and assayed for bioactivity. Liquid chromatography-mass spectrometry (LC-MS) was used to identify peptides that were selected for synthesis and activity testing.

## 2. Results

### 2.1. Infant Characteristics

Thirty-nine infants were enrolled for intestinal sample collection; however, a post-pyloric sampling tube could not be placed in 13 of them due to physiological constraints, and a sufficient sample volume to complete activity testing could not be obtained from another 10 infants. In total, 16 infants were included in the final analysis, from which we obtained 29 intestinal samples. The clinical characteristics for the included infants are shown in Table 1. Infant gestational age at birth ranged from 25 to 41 weeks, and day of life at enrollment ranged from 6 to 57 days.

### 2.2. Growth Effects of Intestinal Peptide Extracts

The infant intestinal peptide extracts were screened for antimicrobial activity on *S. aureus* and *E. coli* and for growth-promoting activity on *B. infantis* after an 18 h incubation. Sterility checks of peptide extracts incubated in sterile Mueller-Hinton broth without bacteria confirmed that none were contaminated during extraction. Optical density readings were compared with a control of bacteria incubated with sterile PBS to determine the percentage change in growth. The results of the screening assays are shown in Figure 1.

Active peptide extracts were identified for only *S. aureus* and *B. infantis*, as no extract inhibited *E. coli* growth to a significant degree. *S. aureus* was the most susceptible to peptide influence on growth, with three of the samples having reduced OD_600_ by >50% at the highest concentration and continuing to be active up to eight-fold dilution, and another two extracts reducing OD_600_ by ~20% for at least one dilution. Twenty-one peptide extracts promoted *S. aureus* growth and only three were inactive at assayed concentrations.

None of the peptide extracts had growth-inhibiting activity against *E. coli*. Nineteen of the extracts promoted *E. coli* growth, and the remaining ten were inactive at all dilutions. For *B. infantis*, six of the extracts had growth-promoting activity. One extract increased OD_600_ by 30% and another three extracts increased it by >20% at either the highest concentration or two-fold dilution. Five extracts had growth-inhibiting activity against B. infantis, and 18 had no activity.

There was no one peptide extract that had simultaneous growth-inhibiting activity against *S. aureus* and growth-promoting activity against *B. infantis* (Figure 2). Extracts 21 and 28 had high inhibitory activity against both *S. aureus* and *B. infantis*, and extracts 12, 26, and 27 promoted the growth of both bacteria. Extracts 1, 4, and 6 had inhibitory activity against *S. aureus* without impacting the growth of *B. infantis*, and similarly, extract 19 promoted *B. infantis* growth without impacting *S. aureus* growth.

### 2.3. Peptide Profiles of the Intestinal Samples

A peptidomic analysis of the intestinal peptide extracts identified 6645 milk peptides, with 5251 derived from human milk proteins, 1233 from bovine milk proteins, and 161 that could come from either due to sequence overlaps. Of the total peptides, 814 had an identical primary sequence to one or more other peptides but with different post-translational modifications. The mean peptide count was 2455.1 ± 727.9 (mean ± standard deviation), the mean abundance was 1.53 × 10^11^ ± 1.47 × 10^11^, and the mean peptide concentration was 4114.8 ± 2374.6 µg/mL.

Peptides were identified from 223 proteins, 160 of which were human milk proteins and 63 of which were bovine milk proteins. The relative percentage of peptides from each protein for each intestinal sample is shown in Figure 2. Most intestinal peptide extracts were primarily composed of human casein peptides except for extracts 1–4 and extract 20, all of which were collected from ostomy output rather than by gravity drip. These extracts had a higher percentage of serum albumin peptides and peptides from human whey or fortifier proteins. These extracts also had a much lower overall peptide abundance than the other intestinal extracts, likely since they spent more time sitting in the ostomy bag at room temperature before collection, allowing for additional protease activity (Appendix A). The intestinal peptide extracts that inhibited *S. aureus* growth had diverse peptide profiles, with extracts 1 and 4 being serum albumin-dense, extract 28 having much larger than average levels of perilipin-2 and polymeric Ig receptor peptides, and extracts 6 and 21 having casein-dense profiles. Conversely, the intestinal peptide extracts that stimulated *B. infantis* growth were all similar to each other and were composed mostly of human β-casein peptides. All of the *B. infantis*-stimulating extracts except for 22 had higher total peptide abundance compared to the other extracts (1.25–2.65 times higher than the average).

### 2.4. Bioactivity of the Candidate Synthetic Peptides

The identified peptides were first compared to the MBPDB to identify the known bioactive peptides in the infant intestine and peptides with highly homologous sequences (≥80% match). From all intestinal samples, there were 73 known bioactive peptides (14 from human milk proteins, 55 from bovine milk proteins, and 4 with shared sequences between human and bovine) and 467 homologous peptides (173 human, 286 bovine, and 8 shared). The sequences and activities of the identified known bioactive peptides are presented in Appendix A.

Based on each peptide’s percentage abundance and Pearson correlation coefficient, the list of 6645 peptides was narrowed down to 18 with potential antimicrobial activity against *S. aureus* and 13 with potential growth-promoting activity for *B. infantis* (Appendix A). From these 31 peptides, 11 were selected for synthesis with the aim of choosing from several milk proteins and different regions within a protein. 

The results of the growth assays for the peptides incubated with the bacteria are shown in Table 2. Of the eleven peptides, MIC values within the range of concentrations tested could be determined for six peptides for both *S. aureus* and *E. coli*. The most active antimicrobial peptides were Peptide 5 from α_s1_-casein and Peptide 11 from serum albumin (Figure 3A). Peptide 5 had an MIC of 2500 μg/mL for both *S. aureus* and *E. coli*, but the first signs of growth inhibition were noticeable at concentrations of 500 and 1000 μg/mL, respectively. Peptide 11 had an MIC of 3000 μg/mL for both bacteria. At 2500 μg/mL, peptide 5 inhibited all new colony formation of *S. aureus* and *E. coli* after 8 h, and peptide 11 inhibited colony formation by ~100-fold (Figure 3B,C).

The only growth-promoting effects of any of the peptides were identified for Peptide 1, Peptide 2 and Peptide 10, which promoted *E. coli* growth at 500, 2500, and 2500 μg/mL, respectively. Despite the activity of the intestinal peptide extracts, none of the synthesized peptides promoted *B. infantis* growth at any concentration. None of the peptides fully inhibited *B. infantis* growth either, although many began to show inhibition (>15% reduction in OD_600_) at concentrations of 2000 µg/mL and up. At concentrations between 500–2000 μg/mL, Peptide 5 and Peptide 11 partially inhibited *S. aureus* and *E. coli* growth and had no effect on *B. infantis* growth, and at concentrations of 2500 and 3000 μg/mL, fully inhibited *S. aureus* and *E. coli* and only partially inhibited *B. infantis*.

## 3. Discussion

Until now, novel bioactive human milk peptides have primarily been identified from undigested milk or in vitro digests of milk [35]. In vitro modeling, however, does not necessarily reflect the range of in vivo biology. Peptides that are released by the proteolytic digestion of milk are not guaranteed to survive further GI digestion, and it is difficult to create in vitro digestion methods that accurately mimic the immature infant GI system [36]. Though previous peptidomic studies have found that several species of bioactive peptides from human milk are released after gastric digestion [37,38], these studies were restricted to identifying only already-known peptides deriving from only a few regions of β-casein, κ-casein, α-lactalbumin, and lactoferrin [38], and were limited to an early stage of digestion. Identifying novel bioactive human milk peptides from infant digesta, as done in the present study, improves on previous procedures by immediately establishing the relevance of these peptides and eliminating the question of whether they are released during infant digestion. The major drawback with identifying peptides from infant digesta is acquiring a sufficient volume of sample with which to perform the necessary screening assays. To overcome this challenge, we assayed the peptide extracts of many individual infant intestinal samples and compared the peptide profiles of those with activity versus those without. Bioactive milk peptides are typically identified through in silico analysis [39] or through iterative fractionation [20,40,41], which requires a large initial sample volume. The present strategy used less than one milliliter of volume to complete. Furthermore, the peptides were assayed at the same concentration as they were found in the intestinal tract, thus providing evidence for the potential health effects of peptide bioactivity inside the infant GI system such as shaping of the gut microbiota. 

The peptide extracts and synthetic peptides were tested against three representative bacterial species: *S. aureus*, *E. coli*, and *B. infantis*. Depending on geographic location, *S. aureus* and *E. coli* are responsible for up to 18% and 23% of cases of neonatal sepsis, respectively [42,43,44]; and high levels of fecal *E. coli* has been associated with necrotizing enterocolitis outbreaks [45]. *B. infantis* is the predominant colonizer of a healthy infant microbiome and is associated with reduced inflammation and dysbiosis [46]. From the 29 intestinal peptide extracts assayed, five inhibited *S. aureus* activity and six promoted *B. infantis* activity. None of the samples had both activities, indicating that a peptide profile that can suppress pathogen colonization may be distinct from one that can promote commensal bacteria colonization. These results demonstrate that even within the small number of infant digestive samples available for this investigation, there was notable variation in the antimicrobial or bifidogenic activity of each patient intestinal peptide extract. This variation could arise from differences intrinsic to each infant, e.g., the protein profile of the feed milk [47,48], protease abundance or activity [49,50], the extent of digestion at the time of sampling [27,51,52], or other factors yet to be discovered. These results are the first confirmation that milk peptides in the intestinal tract have the ability to influence the growth of bacteria.

Human milk contains a variety of intact bioactive factors that protect the infant from enteral infection and promote a healthy gut environment. Secretory IgA is the principal Ig in human milk. Secretory IgA resists GI digestion and prevents enteric infection by binding to bacterial adhesion sites [53] and inhibiting bacterial translocation [54]. Lysozyme increases the abundance of bacteria associated with a healthy gut and decreases those associated with disease [55], and lactoferrin stimulates intestinal cell development, promotes bifidobacteria and lactobacilli growth, and reduces risk of infectious disease [56]. Beyond proteins, human milk oligosaccharides both reduce bacterial adhesion to intestinal cells and are preferentially utilized by bifidobacteria as an energy source [57]. Milk peptides are another facet to the suite of immunological factors provided in human milk that protect the infant from disease, and future work on their activity in vivo is required to elucidate the magnitude of their contribution.

Though all the individual candidate peptides in the present study had some antimicrobial activity at up to 3000 µg/mL for *S. aureus* and seven had activity for *E. coli*, MIC values were determined only for six of the candidate peptides. All six peptides are novel antimicrobials from human milk, though three are related to previously identified peptides. Peptide 4 is derived from the C-terminus of β-casein, a region from where several antimicrobial peptides have been identified [28]. Peptide 7 is a fragment of a previously identified antimicrobial peptide from κ-casein [58], and Peptide 8 is a fragment of human lactoferrampin from lactoferrin [59]. Peptides 5, 9, and 11 are the first antimicrobial peptides to be identified from human α_s1_-casein, osteopontin, and serum albumin, respectively, and their sequences and activities have been added to the MBPDB. However, the MICs of these peptides are fairly weak, on the range of 20–30 times higher than human lactoferricin [20]. 

It is unlikely that the MICs determined for the synthetic peptides were achieved in the intestinal peptide extracts. The mean peptide concentration of the intestinal extracts was around 4 mg/mL, thus even the most active peptides would have to account for more than half of the peptide concentration on their own. Derivatives of these peptides extended or shortened at either the N- or C-terminus might improve the efficacy, as has been shown with lactoferricin [60]. The activity of the peptide extracts may not be due to a high concentration of specific peptides but the accumulated concentration of peptides with similar sequences from the same region of a milk protein, or their individual local concentrations in the digesta and their interactions with bacteria in the gut. Furthermore, it may be that the purpose of antimicrobial milk peptides is not to eliminate bacteria in the infant intestine, like an antibiotic would, but to put non-lethal negative growth pressure on harmful bacteria so that commensal species can flourish. As infants in the NICU receive feeds at a maximum of every three hours [61], their GI system is constantly being replenished with doses of peptides that we have shown can suppress *S. aureus* and *E. coli* growth. In addition to the newly discovered antimicrobial peptides, the intestinal samples also contained 22 previously known antimicrobial peptides: 2 each from human β-casein and bovine κ-casein, 3 from bovine α_s1_-casein, 4 from bovine α_s2_-casein, 5 from bovine β-lactoglobulin, and 6 from bovine β-casein. These peptides may have contributed to the overall activity of the growth-inhibitory extracts for *S. aureus* and *B. infantis*, as each active extract contained multiple known antimicrobial peptides.

Several milk peptides have been discovered to possess bifidogenic activity. Caseinomacropeptide, a large glycopeptide from bovine κ-casein, has stimulating activity for several bifidobacterial species [62,63]. The enhanced growth caused by caseinomacropeptide may be due to its multiple fermentable sugars that bifidobacteria can preferentially use [64]. Bifidogenic activity was also characterized for three peptides from human lactoferrin [32], one from polymeric Ig receptor [32], and one from bovine lactoferrin [65]. These five peptides all contained a disulfide bond and were identified from pepsin hydrolysates of human or bovine milk that was iteratively fractionated and tested. Bifidobacteria have a surface lactoferrin-binding protein that may play a role in recognizing disulfide-bonded peptides to stimulate growth [66]. In the present study, peptide identification was performed with LC-MS conditions that optimized the number of peptides identified but were unable to determine post-translational glycosylation or disulfide bond formation. As none of the unmodified candidate peptides stimulated *B. infantis* growth, it may be that unidentified, modified peptides were responsible for the stimulation caused by the peptide extracts. Lactobacilli are another group of commensal bacteria whose growth can be enhanced by milk peptides [67], though the only single sequences that have been identified with lactobacillogenic activity are bovine and caprine caseinomacropeptide (likely due to their glycosylation residues) [63]. The present study did not test peptides with lactobacilli, but these and other species should remain under consideration for bioactive peptide testing if the microbiome-modulating properties of human milk peptides are to be fully understood.

Though these results showed that peptides in the infant intestine have antimicrobial and bifidogenic properties, it was only partially successful in identifying single bioactive milk peptides that could account for the overall activity of the extracts. The methods utilized can be improved by expanding coverage of the sample peptidomes through identification of peptides outside the optimal size range and peptides with single or multiple post-translational modifications. Further refinements to label-free quantitation or the application of absolute quantitation methods will also indicate which peptide species are truly the most abundant in the samples. In addition, improvements to methods used to select candidate peptides can be made through machine learning algorithms or quantitative structure-activity relationship modeling to the identified peptides. Furthermore, as the majority of the human bacterial population reside in the large intestine, it will be important to determine whether the identified peptides can survive to the large intestine without undergoing additional digestion or whether other peptides with similar activities may be released there.

In conclusion, this paper represents another step in the process of determining the relevance of bioactive human milk peptides in the infant. Though past studies have focused on identifying bioactive peptides from undigested or in vitro digested milk, these peptides may not be present or survive to their sites of activity in the infant GI tract. This is the first study to confirm that infants may release peptides with antimicrobial and bifidogenic activity in the intestinal tract. These peptides may play a role in shaping the local microbiota of the region of the intestine in which the peptides are generated, or may have a more general impact if the peptides persist into the larger intestine. This shaping could have significant effects on infant health and represents how products of protein digestion can benefit the infant beyond provision of precursors for anabolism. Potential applications of these peptides could be as supplements to infant feed to account for altered protein digestion or as inclusions in formula to better mimic the functionality of HM. Future research should investigate factors that may lead to differential peptide release (feed fortification, infant gestational age, digestion time, etc.) and what health-promoting effects these bioactive peptides may have in vivo so that they may be applied to clinical improvements.

## 4. Materials and Methods

### 4.1. Materials

Trichloroacetic acid was obtained from Sigma-Aldrich (St. Louis, MO, USA). HPLC-grade acetonitrile and phosphate-buffered saline (PBS) were obtained from Thermo Fisher Scientific (Waltham, MA, USA), and trifluoroacetic acid and HPLC-grade formic acid were obtained from EMD Millipore (Billerica, MA, USA). Mueller-Hinton broth (MHB) and Bacto Agar were obtained from BD (Franklin Lakes, NJ, USA). Stock bacteria (*Staphylococcus aureus* ATCC 12600, *Escherichia coli* BL21 (DE3), *Bifidobacterium longum* spp. *infantis* ATCC 15697) were obtained from ATCC (Manassas, VA, USA). Candidate peptides were synthesized to ≥98% purity by GenScript (Piscataway, NJ, USA).

### 4.2. Sample Collection

Ethical approval for this cross-sectional study was granted by the Institutional Review Board at Oregon Health and Sciences University (STUDY 00017968, 1 March 2019). Infants were enrolled in the neonatal intensive care unit following informed consent from the parents. For inclusion in this study, infants had to have an indwelling nasogastric or orogastric feeding tube and had to tolerate full enteral feeding volumes. Infants were excluded from the study if they had anatomic or functional GI disorders that would affect protein digestion, were medically unstable, were nonviable, or had disorders that would be expected to affect normal digestion. 

Upon enrollment, a sampling tube was placed into the distal duodenum or proximal jejunum, with the position of the sampling port confirmed by abdominal X-ray. Human milk (either mother’s own milk or pasteurized donor’s milk) with and without fortification (Similac Human Milk Fortifier or Neosure^®^ fortifier) was fed to infants via a nasogastric tube over one hour or less. Sampling was performed continuously over a period of two hours after the initiation of feeding. Samples were collected from the nasoduodenal/jejunal tube via gravity flow as the digesta passed the collection tube port if a post-pyloric tube had been placed, or collected from a jejunostomy bag if present. Intestinal samples were collected into sterile, low-protein binding collection tubes and placed immediately on ice then stored at −80 °C. All samples were transported to Oregon State University on dry ice and stored at −80 °C upon arrival. Infant demographic and anthropometric data were recorded at time of feeding. 

### 4.3. Peptide Extraction

Intestinal samples were thawed on ice; 1 mL of each sample was centrifuged at 14,000× *g* for 10 min at 4 °C to separate fats and solids, and the infranatant was pipetted into a new tube. To ensure complete extraction of the peptides, the remaining fats and solids were agitated with a vortex mixer with 500 mL of nanopure water and recentrifuged at the same speed and time. The second infranatant was added to the previous infranatant. Each sample was mixed with an equivalent volume of 24% trichloroacetic acid and centrifuged at 12,000× *g* for 20 min at 4 °C to precipitate remaining intact proteins. Peptides were separated from the supernatant via C18 solid-phase extraction following our previous methodology [27]. The eluate was freeze-dried and rehydrated in 1 mL of sterilized PBS for bioactivity screening.

### 4.4. Peptide Concentration Determination

The concentrations of the intestinal peptide extracts were determined by the Pierce™ Quantitative Colorimetric Peptide Assay (Thermo Fisher Scientific, Waltham, MA, USA). An aliquot of 20 μL of intestinal fluid was diluted in 80 μL of nanopure water. The samples were mixed with 400 μL of ice-cold ethanol and incubated for 2 h at −20 °C to precipitate intact proteins. Samples were centrifuged at 12,000× *g* for 30 min at 4 °C, and the protein pellet was discarded. Ethanol was removed from the supernatant via SpeedVac. The peptides were reconstituted in 100 μL of water for concentration determination, and the final results of the colorimetric assay were multiplied by the dilution factor of 5.

### 4.5. Mass Spectrometry Analysis

Peptides from 20 µL of each intestinal sample were extracted as described above and dissolved in 20 µL of nanopure water after freeze drying for LC-MS analysis. LC-MS was performed as previously described [34] with the following change: as a number of the infants were fed milk with Neosure^®^ bovine-based fortifier, peptides were identified using Proteome Discoverer 2.2.0.388 with a Sequest HT search against a database that contained both human and bovine milk proteins. Dynamic peptide modifications only included phosphorylation of serine and threonine and oxidation of methionine. 

### 4.6. Intestinal Peptide Extract Bioactivity Screening

The bioactivity of the intestinal peptide extracts was determined via the microdilution method. Antimicrobial activity was screened with *E. coli* and *S. aureus* as representatives of common infant pathogens responsible for infectious disease, and growth-promoting activity was screened with *B. infantis*, one of the ideal colonizers of the infant gut. For the antimicrobial assays, colonies of bacteria were selected and inoculated in 2 mL of MHB and incubated at 37 °C for 24 h. The inoculum was diluted to 2 × 10^5^ CFU/mL with MHB. For the growth-promoting assays, 100 µL of stock bacteria were inoculated in 10 mL of reinforced clostridial broth supplemented with 0.1% ascorbic acid and incubated under anaerobic conditions (BD BBL™ GasPak™) at 37 °C for 24 h. Optical density was measured at 600 nm, and the inoculum was diluted to an optical density of 0.05 with reinforced clostridial broth.

The intestinal peptide extracts were serially diluted with PBS to concentrations of 1×, 1/2×, 1/4×, 1/8×, 1/16×, 1/32×, 1/64×, and 1/128×. In a 96-well plate, 50 µL of each dilution was incubated with 50 µL of inoculum in duplicate, along with a negative control of 50 µL of inoculum with 50 µL of pure PBS and a sterility test of 50 µL of peptide with 50 µL of broth. Growth was determined by optical density readings at 600 nm (OD_600_) taken at 0 h (T_0_) and 18 h (T_18_). The following equation was used to determine percent inhibition or promotion of bacterial growth:

(1)$$100\times \frac{\left({\mathrm{OD}}_{600(\mathrm{Sample})}\mathrm{at}T_{18}-{\mathrm{OD}}_{600(\mathrm{Sample})}\mathrm{at}T_{0}\right)}{\left({\mathrm{OD}}_{600(\mathrm{Control})}\mathrm{at}T_{18}-{\mathrm{OD}}_{600(\mathrm{Control})}\mathrm{at}T_{0}\right)}$$

Peptide extracts were classified as “growth-inhibiting” if they decreased OD_600_ at any dilution, “growth-promoting” if they increased OD_600_, and “inactive” if they did not change OD_600_ for all dilutions. The threshold for activity was set as anything greater than the variation of OD_600_ for the bacteria grown without peptide under the same conditions (~15%).

### 4.7. Candidate Peptide Selection and Bioactivity Determination

The peptidomic data was compared with the Milk Bioactive Peptide Database (MBPDB) [28] to identify known and potential bioactive peptides. The search type was “Sequence,” and a similarity threshold of 80% was used to identify peptides with high sequence homology to known bioactive peptides that may be predictive of bioactivity.

The peptide profiles of active and inactive intestinal samples were compared to identify candidate peptides for synthesis. The percentage abundance of each peptide in a sample was calculated by dividing each peptide’s abundance (the ion intensity of the peptide as measured by the mass spectrometer) by the sample’s total peptide abundance. Pearson correlation coefficients were determined for the effect of each peptide’s percentage abundance within a sample on the sample activity with R version 3.6.1. Candidate peptides were selected from active samples based on high percentage abundance, Pearson correlation coefficient, and ratio of percentage abundance in active samples to percentage abundance in inactive samples.

Candidate peptides were synthesized and dissolved in sterile nanopure water. Antimicrobial and growth-promoting assays were carried out as described above with serial dilutions ranging from 3000 μg/mL to 15.6 μg/mL. Minimum inhibitory concentration (MIC) was determined by the concentration at which all visible growth was inhibited.

## Figures and Tables

**Figure 1 ijms-22-02377-f001:**
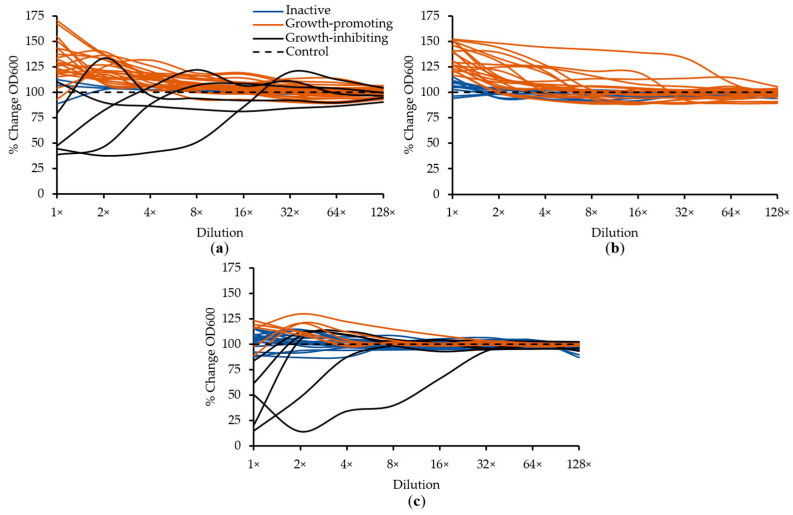
The effect of serial dilutions of intestinal peptide extracts on the OD_600_ of bacteria after an 18 h incubation. Each line represents the mean of an individual peptide extract tested in duplicate. The following bacteria were tested: (**a**) *Staphylococcus aureus*, (**b**) *Escherichia coli* and (**c**) *Bifidobacterium longum* spp. *infantis*.

**Figure 2 ijms-22-02377-f002:**
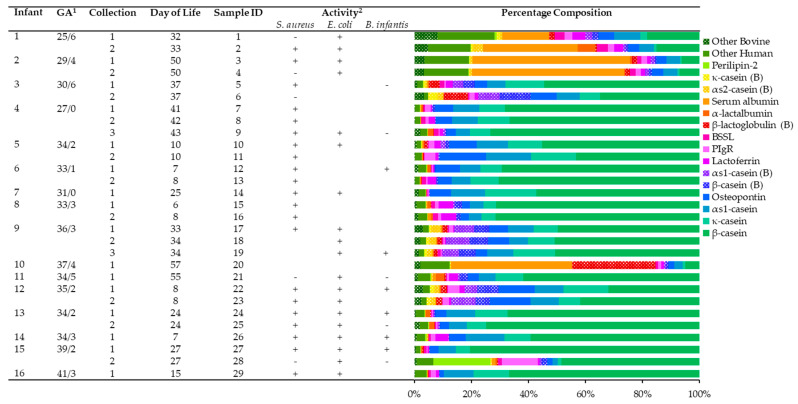
Infant birth information, screening activity, and peptide composition for each intestinal peptide extract. Percentage composition of the peptides from each intestinal sample are sorted by protein from the highest to lowest mean abundance. Bovine milk proteins are labeled with a (B) in the legend and are filled with a dotted pattern. PIgR, polymeric immunoglobulin receptor; BSSL, bile salt-stimulated lipase. ^1^ GA units are w/day. ^2^ + Indicates the sample stimulated growth of the bacteria, - indicates the sample inhibited growth, and no symbol indicates the sample was inactive.

**Figure 3 ijms-22-02377-f003:**
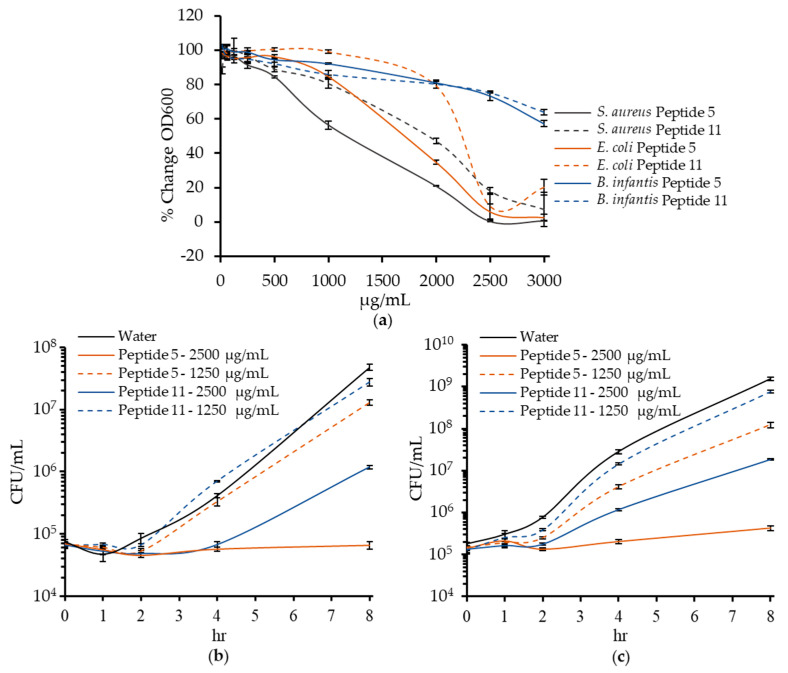
Antimicrobial activity of peptides α_s1_-casein (35–44) and serum albumin (35–44). (**a**) Percentage change in OD_600_ of *Staphylococcus aureus*, *Escherichia coli*, and *Bifidobacterium longum* spp. *infantis* after 18 h. Change in CFU/mL over time of (**b**) *S. aureus* and (**c**) *E. coli* after incubation with peptides. Data are shown as mean ± standard deviation.

**Table 1 ijms-22-02377-t001:** Clinical characteristics of infants from whom intestinal samples were collected.

Characteristic	Value
No. Infants	16
No. Samples	29
No. Preterm	26 (89.7) ^1^
No. Fortified	16 (55.2) ^1^
No. Donor Milk	14 (48.3) ^1^
No. Ostomy	5 (17.2) ^1^
Gestational Age (w)	33.15 ± 4.12 ^2^
Day of Life (d)	28.07 ± 15.96 ^2^
Weight (kg)	2.11 ± 0.51 ^2^
Length (cm)	44.23 ± 3.79 ^2^
Feed Volume (mL)	40.03 ± 13.92 ^2^
Energy Intake (kcal/kg/d)	122.82 ± 26.03 ^2^

^1^ Data in parentheses is the percentages of samples. ^2^ Data are presented as mean ± standard deviation.

**Table 2 ijms-22-02377-t002:** Antimicrobial activity of synthesized human milk peptides.

ID	Sequence	Protein	Position	*S. aureus*	*E. coli*	*B. infantis*
				MIC ^1^	MIC ^1^	MIC ^1^
1	HLPLPLLQPLMQQVPQPI	β-casein	140–157	>3000	>3000	>3000
2	LLNPTHQIYPVTQPLAPVHNPIS	β-casein	203–225	>3000	>3000	>3000
3	HQIYPVTQPL	β-casein	208–217	>3000	>3000	>3000
4	LAPVHNPI	β-casein	217–224	3000	3000	>3000
5	EPIPLESREE	α_s1_-casein	35–44	2500	2500	>3000
6	YANPAVVRPHAQIPQR	κ-casein	81–96	>3000	>3000	>3000
7	RPNLHPS	κ-casein	110–116	3000	3000	>3000
8	EKFGKDKSPKFQ	Lactoferrin	295–306	3000	3000	>3000
9	DMLVVDPK	Osteopontin	283–290	3000	3000	>3000
10	MTSALPIIQK	Perilipin-2	62–71	>3000	>3000	>3000
11	FKDLGEENFK	Serum albumin	35–44	3000	3000	>3000

^1^ MIC units are µg/mL.

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
