# Peer review of "Peptides from the Intestinal Tract of Breast Milk-Fed Infants Have Antimicrobial and Bifidogenic Activity"

_ijms, 2021, doi:10.3390/ijms22052377_

Round 1

Reviewer 1 Report

The present paper entitled “Milk peptides in the intestinal tract of breast milk-fed infants have antimicrobial and bifidogenic activity“ presents original data using biological samples that are nowadays very rare to collect.  The topic is thus of interest for the scientific community. However, first of all, the title is misleading, as this is not clearly established that the milk peptides are responsible of the bifidogenic activity. Other compounds, as discussed in the discussion section, may have been responsible of this activity. In addition, some details not only regarding the material and methods but also regarding the data analysis are further required to strengthen the results and the conclusion. Particularly, the effects of the different confounding factors should be more thoroughly studied (fortification, maturity of the digestive tract…). It is not stated on which type of milk the infants were fed (mother’s own milk or milk from a bank, which has thus be pasteurized?).

Regarding the material and methods :

  • Was there any clinical trial registration, such as on the international database clinicaltrials.gov ?
  • How many infants were enrolled in the study and for whom the study could not be completely achieved?
  • The sampling tube was inserted in addition to the feeding tube, isn’t? How was this tube removed?
  • Feeding time : 30 min or less, what means less?
  • Why the digesta were freeze-dried before being resuspended in the same volume? What could be the impact of such processing? Was the same extract (freeze-dried and resuspended) used for both peptide and bacteria growth used (L 354) ?
  • At what postprandial time was collected the intestinal digesta? This could be a confounding factor that would need to be checked for.
  • The effect of the different confounding factors should be assessed. Particularly, was there any difference between the subgroups of term vs preterm infants, fortified vs. non-fortified milk, postprandial time, … ?
  •  

Results & Discussion:

  • Table 1 : preterm / fortification, what does the counting refer to ?
  • Table 2 and fig 2 should be merged for a better understanding and also as similar information are given. Particularly, information related to the same subject should be easily identified. In addition, would that be possible to average the data collected on the same subject?
  • Was the Bifidobacterium longum infantis representative of the Bifidobacteria species?
  • Was there any quantification of the candidate bioactive peptides in the digesta?
  • As stated L 222, lactoferrin may promote bifidobacterial. Then, L271, you should also state that bioactive proteins could also be responsible of this bacteria growth.

Author Response

The present paper entitled “Milk peptides in the intestinal tract of breast milk-fed infants have antimicrobial and bifidogenic activity“ presents original data using biological samples that are nowadays very rare to collect.  The topic is thus of interest for the scientific community. However, first of all, the title is misleading, as this is not clearly established that the milk peptides are responsible of the bifidogenic activity. Other compounds, as discussed in the discussion section, may have been responsible of this activity. In addition, some details not only regarding the material and methods but also regarding the data analysis are further required to strengthen the results and the conclusion. Particularly, the effects of the different confounding factors should be more thoroughly studied (fortification, maturity of the digestive tract…). It is not stated on which type of milk the infants were fed (mother’s own milk or milk from a bank, which has thus be pasteurized?).

Thank you very much for taking the time to read our manuscript and for your insightful comments. We have changed the wording of the manuscript title to more accurately represent the results conferred within. Furthermore, we absolutely agree that there are numerous factors that could contribute to the release of bioactive peptides. The aim of the present study was not to tease apart those factors (though such a study would be of immense interest!) but simply to determine if there exists any ability for infants to release peptides with measurable bioactivity through their own natural digestive processes. This language was added to the discussion on lines 307-308 and 314-316.

Regarding the material and methods:

  • Was there any clinical trial registration, such as on the international database clinicaltrials.gov?

We don’t believe clinical trial registration is applicable for the present study. The design was observational, sampling a representative population of NICU infants at one time. Infants were not enrolled and randomly assigned to an intervention group. The study design was clarified on line 68 and 330.

  • How many infants were enrolled in the study and for whom the study could not be completely achieved?

We enrolled 39 infants for intestinal sample collection, but only used samples from 16 of the infants in the present study. 13 of the infants could not have post-pyloric tubes placed due to physiological constraints, and another 10 infants did not produce enough sample volume to complete the experimental procedures with. This is a good point that you have brought up, and we have added the above information to the manuscript on lines 75-80.

  • The sampling tube was inserted in addition to the feeding tube, isn’t? How was this tube removed?

After sample was collected by gravity flow, the tube was gently extracted from the infant duodenum so that unimpeded feeding could continue.

  • Feeding time : 30 min or less, what means less?

30 minutes is actually leftover from an earlier draft; the true time limit was one hour. The NICU’s standard procedure was to administer the feed over a specific time period which we set at one hour for this study (infants who could not be fed in under one hour were not included), however some infants received their full feed before an hour was reached due to high tolerance for feeds.

  • Why the digesta were freeze-dried before being resuspended in the same volume? What could be the impact of such processing? Was the same extract (freeze-dried and resuspended) used for both peptide and bacteria growth used (L 354) ?

The digesta were freeze-dried to eliminate the 80% acetonitrile solvent after C18 extraction, which could have profoundly influenced bacterial growth and the ability of the peptides to bind to the column during LC-MS/MS. The extraction procedures for peptides incubated with the bacteria and peptides injected into the mass spectrometer for identification were identical. Freeze-drying is routintely performed for such analyses, and we expect it to have little influence on the peptides’ composition.

  • At what postprandial time was collected the intestinal digesta? This could be a confounding factor that would need to be checked for.

Intestinal sampling was initiated when feeding was and was performed continuously over a period of two hours, although the first samples typically weren’t able to be collected until about 15 minutes after the start of feeding. All duodenal fluid available was collected over this time period. This information was added to lines 341-342.

  • The effect of the different confounding factors should be assessed. Particularly, was there any difference between the subgroups of term vs preterm infants, fortified vs. non-fortified milk, postprandial time, … ?

This study was not powered to perform such comparisons between groups of infants and account for confounding factors. The design of the study was not to compare bioactive peptides in the intestine between infants, but to identify if bioactive peptides are even present in the intestinal tract (where they are most relevant to the infant). However, what you propose is important work for the future, and this study will serve as the basis for investigating such differences as we expect that information to be critical to meeting the health needs of the preterm infants.

Results & Discussion:

  • Table 1 : preterm / fortification, what does the counting refer to ?

The count refers to the number of samples that are from preterm infants/infants fortified with a bovine milk-based fortifier. This was clarified a little better in the table.

  • Table 2 and fig 2 should be merged for a better understanding and also as similar information are given. Particularly, information related to the same subject should be easily identified. In addition, would that be possible to average the data collected on the same subject?

We have combined Table 2 and Fig 2 per your suggestion. We included a similar chart that is an average of the data from the same subject as Supplementary Fig X, though the focus of the main paper is on each sample rather than each infant.

  • Was the Bifidobacterium longum infantis representative of the Bifidobacteria species?

Yes, B. infantis is representative of Bifidobacteria. This information was added on lines 207-212 and 379-381.

  • Was there any quantification of the candidate bioactive peptides in the digesta?

Unfortunately we were unable to quantify peptides in the digesta, though this is something that we intend to do going forward as concentration is a key determinant for activity.

  • As stated L 222, lactoferrin may promote bifidobacterial. Then, L271, you should also state that bioactive proteins could also be responsible of this bacteria growth.

Intact proteins were precipitated out with TCA prior to incubation with bacteria to avoid conflating the effect of bioactive proteins and bioactive peptides, so it is unlikely the bifidogenic effect of the extracts shown was due to bioactive proteins.

Reviewer 2 Report

This is an interesting study examining the role of intestinal milk peptide-rich digests upon bacterial growth. These are the comments regarding this manuscript:

  1. How and why did the authors choose the 3 bacteria to be tested? It would have been better to test these digests against a wider range of pathogenic bacteria (plus B. bifidum as a control) implicated in the causation of necrotizing enterocolitis.
  2. The authors should clearly state that while their samples were collected from the small intestine, it is the large intestine where the bulk of GI bacteria reside. As such, there exists the possibility that these digests would be further modified before reaching the gut bacteria.
  3. Data on the infants is lacking. At the very least, how these infants were selected, their sex/ethnic background, feeding history (human versus cow's milk and/or infant formula use), medical condition/s should have been described.
  4. The number of subjects and samples is inadequate. Why and how it was decided to choose these numbers?
  5. It is surprising to include both term (2 infants and 3 samples) together with preterm ones. It is strongly recommended to remove the term subject samples and replace these with more preterm ones, to make the study relevant to the target population.
  6. What are the approximate concentrations of milk peptides in these digests? How does it compare to the levels observed with synthetic peptides, especially the MIC values?
  7. It is surprising that the authors did not test the effects of their samples (and the synthetic peptides) on Lactobacilli, a common (and generally beneficial) gut bacteria.
  8. The authors should discuss the significance of their findings in the context of future potential therapeutic approaches. Do the authors consider administration of high doses of synthetic peptides a viable strategy to prevent growth of pathogenic bacteria? 

Author Response

This is an interesting study examining the role of intestinal milk peptide-rich digests upon bacterial growth. These are the comments regarding this manuscript:

  1. How and why did the authors choose the 3 bacteria to be tested? It would have been better to test these digests against a wider range of pathogenic bacteria (plus B. bifidum as a control) implicated in the causation of necrotizing enterocolitis.

Thank you very much for your comments, we appreciate the time you’ve taken for such in depth edits. S. aureus and E. coli were chosen as they are common infant pathogens responsible for infection, and B. infantis was chosen as it is considered the ideal colonizer of the infant gut. This information was included on lines 213-218 and 363-366. We understand that just because a peptide is active against these bacteria does not mean it is active against the range of pathogens that can lead to NEC or sepsis. Given the complicated etiologies of NEC and neonatal sepsis, the testing of single peptides with single bacteria can only provide so much information before new study designs are required. However, these results can still serve as a foundation for future investigations into the association between bioactive peptides and infectious disease.

  1. The authors should clearly state that while their samples were collected from the small intestine, it is the large intestine where the bulk of GI bacteria reside. As such, there exists the possibility that these digests would be further modified before reaching the gut bacteria.

This information was added on lines 298-302.

  1. Data on the infants is lacking. At the very least, how these infants were selected, their sex/ethnic background, feeding history (human versus cow's milk and/or infant formula use), medical condition/s should have been described.

The infants were selected based on providing enough intestinal fluid to perform the procedures, which was added on lines 75-80. More information on types of feed was added to Table 1 (all infants were fed human milk – either donor milk or mother’s milk), but unfortunately we did not collect sex or ethnicity information.

  1. The number of subjects and samples is inadequate. Why and how it was decided to choose these numbers?

We added more information on enrolling the infants on lines 75-80. We initially enrolled 39 infants, but only 16 provided sufficient sample volume for activity testing. However, our main goal was not to compare between infant groups (in which case, our study would have been underpowered), but to identify bioactive peptides in the infant intestinal tract. As even with our limited sample size, we found several samples that had active intestinal extracts for S. aureus and B. infantis, we were successful with our chosen study design and sample size. We believe the data we showed from the sample population supported our final conclusions, which was that antimicrobial and bifidogenic peptides can be released in the infant intestinal tract.

  1. It is surprising to include both term (2 infants and 3 samples) together with preterm ones. It is strongly recommended to remove the term subject samples and replace these with more preterm ones, to make the study relevant to the target population.

Since the term samples were included in the comparison between active and inactive extracts to determine candidate peptides for synthesis, their inclusion in the paper is necessary. We also have no other preterm samples to replace the term samples with. However, because we draw no conclusions between the preterm and term samples, we believe it is acceptable to keep them in the analysis and keep the focus on bioactive peptide release in infants regardless of gestational age rather than narrowing in on any specific subpopulation (though such work should and will be performed in the future).

  1. What are the approximate concentrations of milk peptides in these digests? How does it compare to the levels observed with synthetic peptides, especially the MIC values?

This is a good point to include. We performed a BCA assay and added information on the concentration of the peptides on lines 121-122, 248-250, and 356-364.

  1. It is surprising that the authors did not test the effects of their samples (and the synthetic peptides) on Lactobacilli, a common (and generally beneficial) gut bacteria.

While lactobacilli are another important group of commensals, they make up a much smaller percentage of the microbiome composition in early infancy than bifidobacteria. Human milk peptides have also previously been shown to be bifidogenic, while none have shown to be lactobacillogenic, although there could be potential for such activity. Thus, it made sense to start with just B. infantis. We included some discussion on the potential of lactobacilli-stimulating peptides on lines 282-288, and will likely pursue this line of research in the future.

  1. The authors should discuss the significance of their findings in the context of future potential therapeutic approaches. Do the authors consider administration of high doses of synthetic peptides a viable strategy to prevent growth of pathogenic bacteria? 

It is difficult to discuss peptides as therapeutics as much more work still needs to be done to clarify and enhance their efficacy, metabolism, absorption, etc, since they are so readily digestible. Instead, the focus of this paper is more on understanding the natural benefits of human milk to the infant in terms of its peptides and applying those benefits to formula or human milk feedings to make up for reduced digestibility or altered protein composition. This discussion was included on lines 312-314.

Reviewer 3 Report

In this manuscript, authors analyzed infants’ intestinal samples, collected after feeding. Extracted and serially diluted milk peptides were analyzed for bioactivity using growth of different bacteria strains as a readout assay.  The following peptidomics approach allowed to identify a range of bioactive peptides, some already known, but also several novel bioactive motifs. Synthetic forms of these identified sequences displayed relatively weak antimicrobial potential compared to human lactoferricin. The performed work is very important for our understanding of nature and composition of human milk-derived anti-bacterial peptides. The manuscript is well written and obtained results are support the conclusion.

Minor comment:

Fig. 3: Several lines in figures a, b, and c have quite pale color  - could it be improved?

Author Response

In this manuscript, authors analyzed infants’ intestinal samples, collected after feeding. Extracted and serially diluted milk peptides were analyzed for bioactivity using growth of different bacteria strains as a readout assay.  The following peptidomics approach allowed to identify a range of bioactive peptides, some already known, but also several novel bioactive motifs. Synthetic forms of these identified sequences displayed relatively weak antimicrobial potential compared to human lactoferricin. The performed work is very important for our understanding of nature and composition of human milk-derived anti-bacterial peptides. The manuscript is well written and obtained results are support the conclusion.

Minor comment:

Fig. 3: Several lines in figures a, b, and c have quite pale color  - could it be improved?

Thank you very much for your comments, we have changed the figure colors for improved legibility and hope you find it much easier to read.

Round 2

Reviewer 1 Report

The revision of the paper has been well undertaken, except for two points :

  • the question regarding whether this is human milk from the own mother or from a human milk bank and regarding the heat treatment applied has not been answered.
  • I do not agree that this is an observational study, although I understand that no feeding intervention was performed.

Author Response

The revision of the paper has been well undertaken, except for two points :

The question regarding whether this is human milk from the own mother or from a human milk bank and regarding the heat treatment applied has not been answered.

We apologize for not mentioning this in our previous response, but we clarified in Table 1 that of the 29 intestinal samples tested, 14/29 were from donor milk feedings, and 16/29 were fortified. The donor milk was pasteurized by the milk bank (information now added to Line 341). The study was not powered to examine differences between donor and mother’s milk, though we plan to examine this in a future study.

I do not agree that this is an observational study, although I understand that no feeding intervention was performed.

We did not register this as a clinical trial as we do not believe it meets the definition of a clinical trial and is more observational in nature. We have published work from this study previously without any clinical trial registration (1-3).

  1. Lueangsakulthai J, Kim BJ, Demers-Mathieu V, Sah BNP, Woo Y, Olyaei A, Aloia M, O'Connor A, Scottoline BP, Dallas DC. Effect of digestion on stability of palivizumab IgG1 in the infant gastrointestinal tract. Pediatr Res. 2020.
  2. Lueangsakulthai J, Sah BNP, Scottoline BP, Dallas DC. Survival of recombinant monoclonal and naturally-occurring human milk immunoglobulins A and G specific to respiratory syncytial virus F protein across simulated human infant gastrointestinal digestion. J Funct Foods. 2020;73:104115.
  3. Kim BJ, Lueangsakulthai J, Sah BNP, Scottoline B, Dallas DC. Quantitative Analysis of Antibody Survival across the Infant Digestive Tract Using Mass Spectrometry with Parallel Reaction Monitoring. Foods. 2020;9(6).

Reviewer 2 Report

While the authors have successfully addressed most of the concerns identified, it is still not justified why term and preterm infants samples are to be considered together. It would be preferable to replace the term samples with appropriate preterm ones and focus on the health implications in preterm subjects only.

Author Response

While the authors have successfully addressed most of the concerns identified, it is still not justified why term and preterm infants samples are to be considered together. It would be preferable to replace the term samples with appropriate preterm ones and focus on the health implications in preterm subjects only.

We included both term and preterm infants so that we could examine the broadest array of digestion within the NICU. Though the study is not powered to compare digestion across these groups, use of these samples was valuable for identifying peptides with potential antimicrobial and bifidogenic action.